# Influence of Pressure on Gas/Liquid Interfacial Area in a Tray Column

**Adel Almoslh \*, Falah Alobaid, Christian Heinze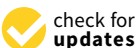 and Bernd Epple**

Institut Energiesysteme und Energietechnik, Technische Universität Darmstadt, Otto-Berndt-Straße 2, 64287 Darmstadt, Germany; falah.alobaid@tu-darmstadt.de (F.A.); christian.heinze@tu-darmstadt.de (C.H.); bernd.epple@tu-darmstadt.de (B.E.)

\* Correspondence: adel.almoslh@tu-darmstadt.de; Tel.: +49-6151/16-23004; Fax: +49-(0)-6151/16-22690

**Abstract:** The influence of pressure on the gas/liquid interfacial area is investigated in the pressure range of 0.2–0.3 MPa by using a tray column test rig. A simulated waste gas, which consisted of 30% $CO_2$ and 70% air, was used in this study. Distilled water was employed as an absorbent. The temperature of the inlet water was 19 °C. The inlet volumetric flow rate of water was 0.17 $m^3$/h. Two series of experiments were performed; the first series was performed at inlet gas flow rate 15 $Nm^3$/h, whereas the second series was at 20 $Nm^3$/h of inlet gas flow rate. The results showed that the gas/liquid interfacial area decreases when the total pressure is increased. The effect of pressure on the gas/liquid interfacial area at high inlet volumetric gas flow rates is more significant than at low inlet volumetric gas flow rates. The authors studied the effect of decreasing the interfacial area on the performance of a tray column for $CO_2$ capture.

**Keywords:** $CO_2$ capture; $CO_2$ absorption; process simulation; validation study; experimental study

## 1. Introduction

The global atmospheric concentration of carbon dioxide and its contribution to global warming requires technical measures to limit the emission of this greenhouse gas. One possibility is by separating the carbon dioxide from the waste gas stream of fossil power plants by applying absorption technologies. The absorption process of $CO_2$ can be performed using solvents with different additives. Amine solvents such as monoethanolamine (MEA), diethanolamine (DEA), and methyldiethanolamine (MDEA) are common for $CO_2$ capture (Wilk et al. 2017) [1]. The reversible reactions and the moderate reactivity between $CO_2$ and amine solutions enable efficient $CO_2$ capture (Yamada 2016) [2]. Barzagli et al. (2014) [3] studied nonaqueous amine solvents such as 2-(isopropylamino)ethanol (IPMEA), 2-(tert-butylamino)ethanol (TBMEA) and N-methyl-2,2'-iminodiethanol(MDEA) for $CO_2$ capture and found that $CO_2$ removal efficiency was in the range of 87–95% at equilibrium, depending on the operational conditions. Although amine absorption processes are widely used for $CO_2$ capture, they have some disadvantages related to equipment corrosion, amine degradation (Sanna 2014) [4], and high energy consumption (Wilk et al. 2017) [1]. The absorption technology for $CO_2$ capture consists mainly of the absorber column and the regeneration unit. The absorber column can be a tray or packed column. The absorbent enters the absorber from the top, and the waste gas, which contains $CO_2$, enters the absorber from the bottom. The gas and the liquid phases contact each other on the trays or the packing materials. The trays or the packed material enhance the gas/liquid interfacial area, which increases the mass and heat transfer between the contact phases. The $CO_2$ component transforms from the gas phase to a liquid phase and then is absorbed.

*Studying the Effect of Pressure on Gas/Liquid Interfacial Area*

Literature reviews reveal that there are various studies concerning the influence of pressure on the hydrodynamics and mass transfer in gas/liquid systems. Some of the studies are devoted to the influence of pressure on the creation of bubbles. These studies used a capillary tube or single orifices connected to a gas chamber. Other studies have investigated the influence of pressure on interfacial areas in a bubble column, packed column, or tray column. Kling et al. (1962) [5] were the first to observe that an increase in pressure at a single gas inlet orifice and constant superficial gas velocity creates a decrease in the initial bubble volume (Oyevaar 1989) [6]. Kling et al. (1962) [5] suggested that the increase in energy content causes the gas to enter further into the liquid, causing elongated bubbles, which separate more easily from the orifice, leading to smaller bubbles at higher pressures. LaNauze et al. (1974) [7] studied the influence of pressure and gas flow rates on the creation of $CO_2$ bubbles in the water at different diameters of orifices photographically. They published the results of the behavior of bubble volume over pressure up to 2.1 MPa at different gas flow rates. They found that the bubble volume is increased when the gas flow rate is increased. Furthermore, it was shown that the bubble volume decreased significantly when the pressure was increased between 0.1–1 MPa, whereas it slightly decreased when the pressure was increased between 1–2.1 MPa. Bier et al. (1978) [8] studied the influence of operating pressure on an initial bubble volume, by sparging $N_2$ or He through a capillary tube into the water or ethanol. The authors concluded that the influence of the operating pressure is much smaller compared with sparging the gas through an orifice connected to a gas chamber since the gas chamber limits pressure vibrations that happen in close gas supply lines (Oyevaar 1989) [6]. Idogawa et al. (1987) [9] noted that the diameter of the initial bubble decreases to 25% when pressure is increased from 0.1 to 15 MPa. Oyevaar et al. (1989) [10] determined interfacial areas at pressures up to 1.85 MPa in a bubble column and a packed column. The authors found that the interfacial areas are unaffected by pressure in the packed bubble column, but that the influence of the pressure on the interfacial areas in the bubble column arises from the generation of smaller bubbles at the gas distributor. Badssi et al. (1988) [11] investigated the effect of the pressure and superficial velocity of gas and liquid on the interfacial area in a laboratory column equipped with cross-flow sieve trays; they checked that each tested variable has an independent influence on the interfacial area. They investigated the effect of the pressure on the total interfacial area in two different gas–liquid systems, $CO_2$-DEA and $CO_2$-NaOH. They reviewed that the total interfacial area decreases when the pressure is increased. Benadda et al. (1996) [12] studied the effect of pressure on the interfacial area in a counter-current packed column. Their experiment conditions were conducted at specific gas mass flow rates of 0.1 kg/m$^2$s, and specific liquid mass flow rates of 5.52 kg/m$^2$s. They concluded that the interfacial area decreases when the pressure is increased between 0.1 and 1.2 MPa. Molga et al. (1996) [13] determined the gas–liquid interfacial during $CO_2$ absorption by using DEA as well as DEA-ETG aqueous solutions. Their experiment device was a bubble column reactor with an inner diameter of 156 mm. The results obtained by Molga et al. (1996) [13] are different; they reviewed that there is no observed influence of pressure on the measured interfacial areas.

Most available studies in the literature are interested in the proportional correlation between the pressure and the absorbed amount of gas in a liquid. In contrast, studies on the effect of pressure on the gas/liquid interfacial area are still limited. One can summarize the objectives of this study as follows:

(1)　To experimentally investigate the effect of the pressure on gas/liquid interfacial area and its effect on the performance of the absorber; an absorber test rig was constructed and operated.

(2)　To assemble a property package for a rate-based model by applying Aspen Plus software for simulation of $CO_2$ absorption using water as an absorbent.

(3)　To validate the mathematical model against experimental data at different operation points.

(4)　To investigate the effect of pressure on the gas/liquid interfacial area for mass transfer and its impact on the performance of a sieve tray absorber for capturing $CO_2$ by using water as an absorbent.

## 2. Experimental

### 2.1. Test Rig Setup

Figures 1 and 2 illustrate an absorber test rig at Technische Universität Darmstadt. The absorber test rig consists of four main parts: a gas mixing unit, an absorber column, a regeneration unit, and a gas analysis unit. The mixing unit consists of two lines, one is connected with a compressed-air source, and the other is combined with a $CO_2$ gas cylinder. The mixing unit is connected to a manifold upstream of the absorber. The absorber is made of a glass column that has a height of 1500 mm, and its internal diameter is 152 mm. The column has 12 glass nozzles to which metal flanges can be attached, ten nozzles used to measure pressures and temperatures in the absorber, and two nozzles for introducing the inlet gas and liquid to the absorber. At the bottom and the top, it is closed by suited metal flanges as well. The top flange contains the exit of the gas, and the bottom flange contains the exit of liquid. Five sieve trays are fixed by threaded rods and inserted inside the absorber. The diameter of the tray is 150 mm, the diameter of the hole is 2 mm, the fraction of the sieve hole area to active area is 0.071, and the weir height is 15 mm. The tray spacing is 240 mm, and the space between the tray and glass wall is closed with rubber seals. A gas analysis unit is connected at the gas outlet line to measure the volumetric fraction of $CO_2$ at the outlet of the absorber.

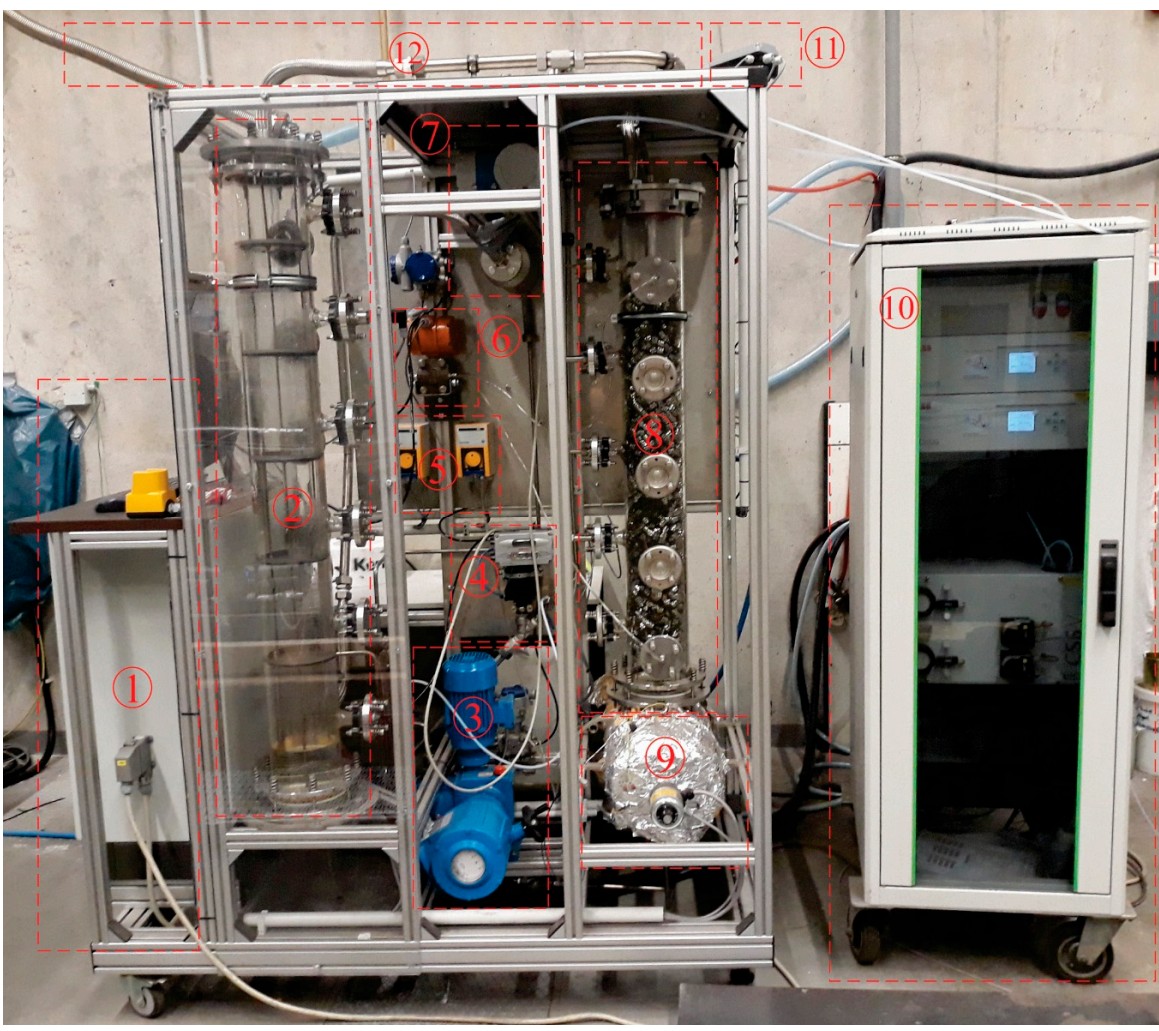

**Figure 1.** Side view of the absorber test rig: 1, control panel; 2, absorber column; 3, recycling pump; 4, liquid level control valve; 5, make-up pump; 6, pressure difference transmitter; 7, Coriolis device; 8, packed column; 9, reboiler; 10, gas analysis unit; 11, pressure control valve; 12, gas outlet.

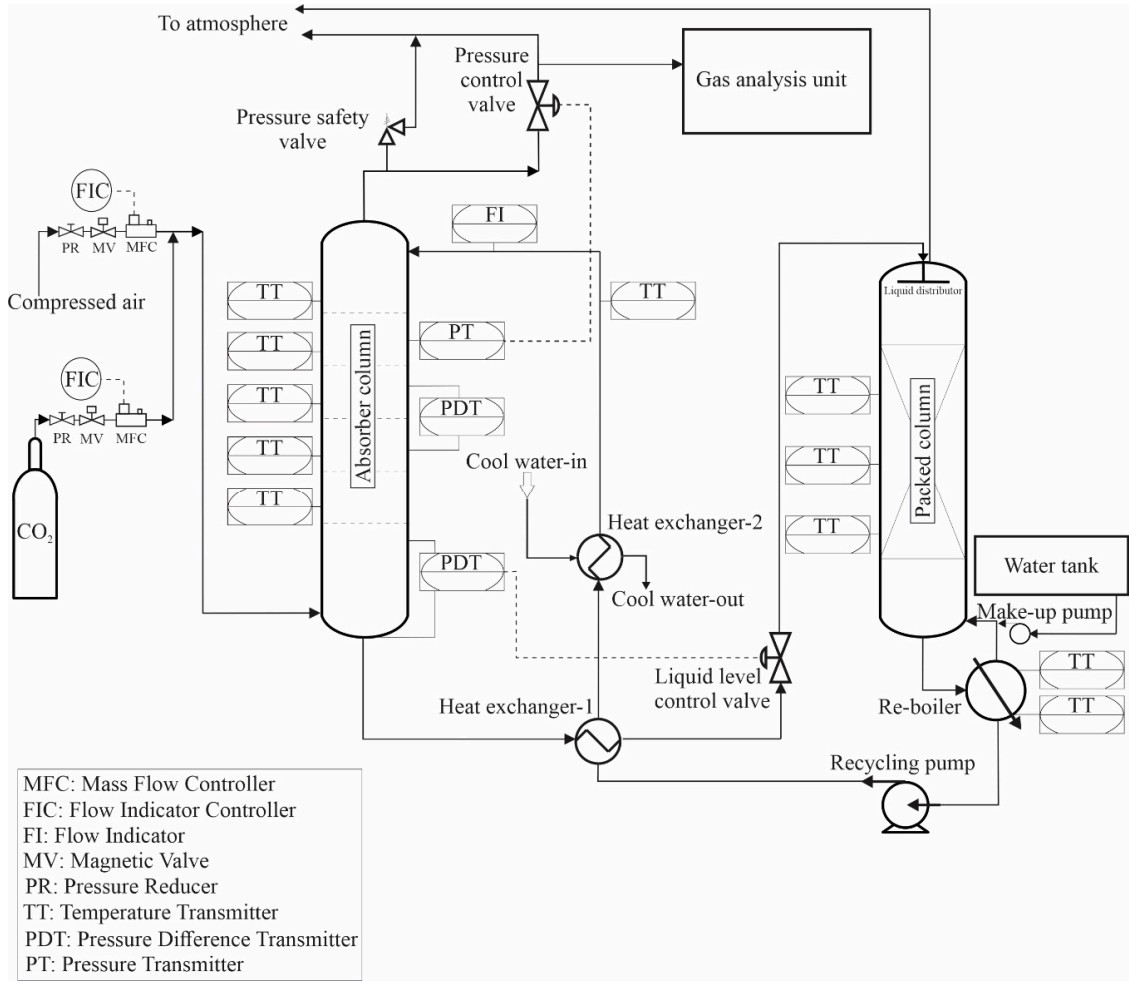

**Figure 2.** Schematic diagram of the absorber test rig.

The objective of the absorbent regeneration unit is to regenerate the absorbent and recycle it to the absorber as a lean absorbent. It consists of two heat exchangers, a reboiler, a packed column, a make-up pump, and a recycle pump. The packed column is a glass column with a diameter of 152 mm and a high of 1300 mm, which is filled to a height of 1 m with a metal packing material from type Pall-Ring 15 mm, with a specific surface of 360 $m^2/m^3$, and free volume 95 %. The rich absorbent enters the packed column through a liquid distributor on the top of the packed column. The purpose of the liquid distributor is to spray the absorbent uniformly on the top of the packings; the manufactured liquid distributor is from a spray type that contains 10 holes distributed uniformly on the liquid distributor. The packed column is placed on the reboiler, which has a thermal power of 4.5 kW. The recycle pump is connected directly to the reboiler, which pumps the water from the reboiler to the absorber. The lean hot absorbent is precooled in the first heat exchanger by exchanging heating with the absorbent, which comes out from the absorber. After that, the precooled absorbent cools down through the second heat exchanger, which is installed before the absorber, by exchanging heating with cool water.

*2.2. Instrumentation and Control Equipment of the Test Rig*

The test rig is provided with various devices and control circuits, which aim at measuring the essential parameters of the absorption process, as well as safe operation. A pressure reducer is fixed on every line of the gas mixer, adjusting the maximum pressure of gas entering the absorber. A magnetic valve is fixed after a pressure reducer, enabling the possibility of opening and closing the gas supply. A mass flow controller (MFC) is used to control the volumetric flow rate for every gas entering the

absorber. A temperature sensor is fixed at each absorber tray to measure the temperature of the fluid on each tray. A Coriolis device and temperature sensor are attached at the inlet of the liquid to measure the inlet flow rate of water as well as the temperature of the inlet water. To estimate the pressure drop due to a tray and liquid holdup, the pressure difference before and after the tray is determined. For this purpose, a pressure difference device, illustrated in Figure 2, is fixed at the absorber.

Five control circles control the test rig. The first control circuit is aimed at controlling the pressure to the set point so that it does not exceed 0.6 MPa (permissible internal pressure of the glass absorber). The pressure control circuit consists of a pressure sensor and a control valve. The pressure sensor is fixed at the top of the tray column, whereas the control valve is installed at the absorber's gas outlet. The pressure control circuit starts controlling the pressure after the gas enters the absorber, resulting in a pressure increase. The pressure sensor sends a signal with the actual value of the pressure to a PID controller. The PID controller compares the set point of pressure and the actual value of pressure and sends a signal to the control valve to open or close, maintaining the pressure at its set point. For safety reasons, in particular, a safety pressure valve at the outlet of the absorber is installed, releasing the pressure inside the glass absorber when it exceeds the value of 0.45 MPa. By this procedure, the glass absorber is protected from any sudden increase of pressure above 0.45 MPa.

The second control circuit controls the liquid level at the bottom part of the absorber to its set point. Controlling the liquid level is essential since it prevents the gas from exiting from the liquid outlet and prevents the accumulation of the liquid inside the absorber to a high level. Without a level controller, the accumulated liquid causes the closing of the inlet of the gas or the possibility of immersing the trays of the absorber with the liquid, which leads to a high-pressure drop and low efficiency of the process. The liquid level control circuit consists of a pressure difference measurement and a control valve. The pressure difference is installed at the bottom of the tray column, whereas the control valve is arranged at the liquid outlet of the absorber. The third control circuit adjusts the level of absorbent inside the reboiler to its set point since there is an absorbent loss due to the absorbent being exposed to stripping and evaporation during the operation. The control circuit consists of a level sensor and a make-up pump. If the level of absorbent decreases below the set point, the level sensor sends a signal to the make-up pump to supply a new absorbent inside the reboiler for the compensation of absorbent loss.

For regeneration of the absorbent, heat is required to increase the absorbent temperature, breaking the bond between the $CO_2$ and the absorbent. For this purpose, a fourth control circuit is used to control the absorbent's temperature inside the reboiler. This control circuit consists of a heater and two temperature sensors that are installed at the top and the bottom of the reboiler. The temperature sensors send signals to the controller that commands the heater. The controller compares the actual value of the temperature inside the reboiler and the set point and sends a signal to the heater to heat the absorbent if the temperature of the absorbent is below the set point. In some cases, for example, the level control circuit does not work correctly. As a result, the level of absorbent inside the reboiler decreases, and it may damage the heater inserted in the reboiler. As a safety procedure, a control circuit (the fifth control circuit) is installed at the reboiler, which aims to shut-down the heater when the liquid level inside the reboiler exceeds the set point of the absorbent level. Accordingly, the heater is protected from any sudden decrease in the absorbent level during the operation.

*2.3. Test Procedure*

The $CO_2$ gas was mixed with air by the gas mixing unit; the air plays a role as a carrier gas. The volume fraction of $CO_2$ was 0.3 in all the experiments. Table 1 illustrates the operating conditions of experiments that have been performed in our study. Two separate experimental series were conducted to investigate the influence of pressure on the gas/liquid interfacial area. The first series of experiments was performed at the inlet gas flow rate of 15 Nm$^3$/h, the second series of experiments were run at the inlet gas flow rate of 20 Nm$^3$/h. In a series of experiments, the pressure was varied in the ranges of 0.2–0.3 MPa. The volumetric flow rate of the inlet water was almost constant at 0.17 m$^3$/h, and its

temperature was controlled at 19 °C. The duration of every experiment was 15 min. The regeneration unit was operated with thermal power of 4.5 kW over time.

**Table 1.** The operation conditions of performed experiments in this study with a $CO_2$ volume fraction of 0.3, inlet water volumetric flow rate of 0.17 $m^3$/h, and inlet temperature of the inlet water of 19 °C.

| Number of the Experiment | The Total Inlet Gas Flow Rate ($Nm^3$/h) | Pressure (MPa) |
| --- | --- | --- |
| 1 | 15 | 0.2 |
| 2 | 15 | 0.21 |
| 3 | 15 | 0.22 |
| 4 | 15 | 0.23 |
| 5 | 15 | 0.24 |
| 6 | 15 | 0.25 |
| 7 | 15 | 0.26 |
| 8 | 15 | 0.27 |
| 9 | 15 | 0.28 |
| 10 | 15 | 0.29 |
| 11 | 15 | 0.3 |
| 12 | 20 | 0.2 |
| 13 | 20 | 0.21 |
| 14 | 20 | 0.22 |
| 15 | 20 | 0.23 |
| 16 | 20 | 0.24 |
| 17 | 20 | 0.25 |
| 18 | 20 | 0.26 |
| 19 | 20 | 0.27 |
| 20 | 20 | 0.28 |
| 21 | 20 | 0.29 |
| 22 | 20 | 0.3 |

## 3. Modeling Approach for Tray Column

### 3.1. Rate-Based Modelling for $CO_2$ Absorption

The two-film theory developed by Lewis and Whitman (1924) [14] explains the mass transfer between gas and liquid. The two-film theory is based on the hypothesis that the gas and liquid phases form a thin layer of fluid on each side of the interface (Whitman 1962) [15]. Figure 3 illustrates the situation of the concentration of component *i* (in our study, *i* represent $CO_2$) at gas/liquid interfacial area. It should be noted that the concentration of component *i* decreases from $C_{i,G}$ in gas bulk to $C_{i,G*}$ in the interface. The difference of concentration creates a driving force for component *i* to shift from the gas bulk to gas film and then from the gas film to liquid film. The growth of component *i* in the liquid film forms a concentration difference between the liquid film and the liquid bulk; likewise, this concentration difference generates a driving force for component *i* to shift it from liquid film to liquid bulk (Whitman 1962) [15].

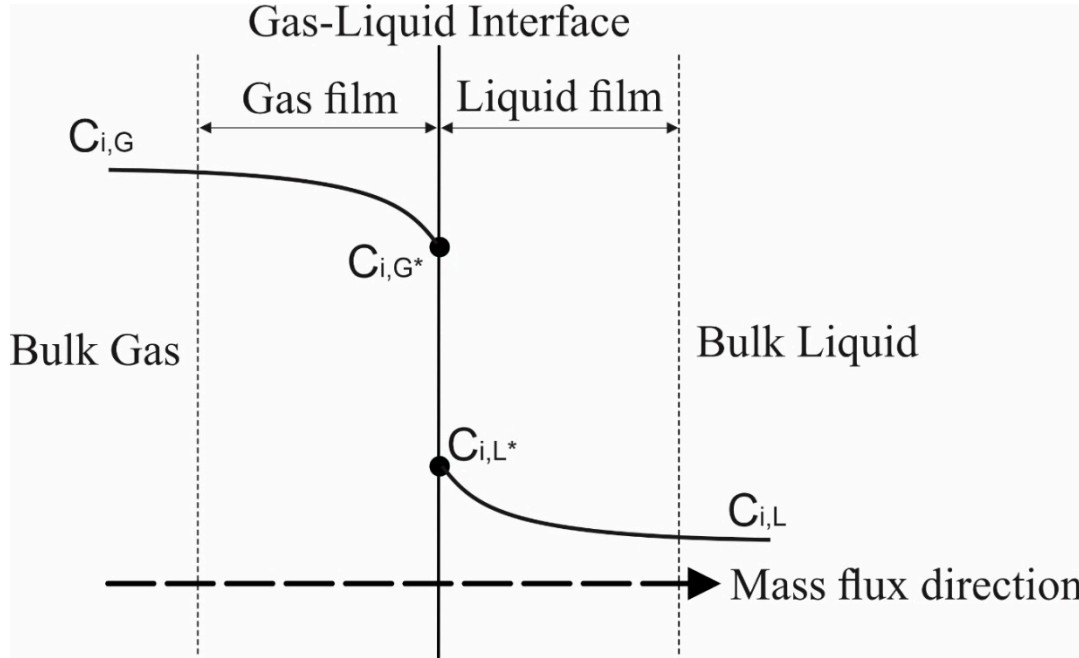

**Figure 3.** Illustration for the situation of the concentration of component *i* at the gas/liquid interfacial area (Doran 2013) (with permission from [16], Copyright Elsevier, 2013).

The rate of mass transfer of component *i* through the gas boundary layer is calculated as follows (Doran 2013) [16]:

$$N_{i,G} = k_G a^* (C_{iG} - C_{iG*}) \tag{1}$$

The rate of mass transfer of component *i* through the liquid boundary layer is calculated as follows (Doran 2013) [16]:

$$N_{i,L} = k_L a^* (C_{iL*} - C_{iL}) \tag{2}$$

where $k_G$, $k_L$ are the mass transfer coefficient of the gas-phase and liquid-phase, respectively, $a^*$ is the interfacial area.

At steady state, the flux of component *i* from bulk gas to the interface must be equal to the flux of component *i* from the interface to the bulk liquid (Ngo 2013) [17]:

$$N_{i,G} = N_{i,L} \tag{3}$$

For modeling the tray column, the rate-based model developed by Pandya (1983) [18] was applied. The rate-based model consists of a set of correlations that calculate the mass and energy transfer across the interfacial area using mass transfer coefficients (Afkhamipour and Mofarahi 2013) [19]. By applying the rate-based model, the absorber column is divided into stages. Figure 4 shows a stage of the column, which represents a tray in the column.

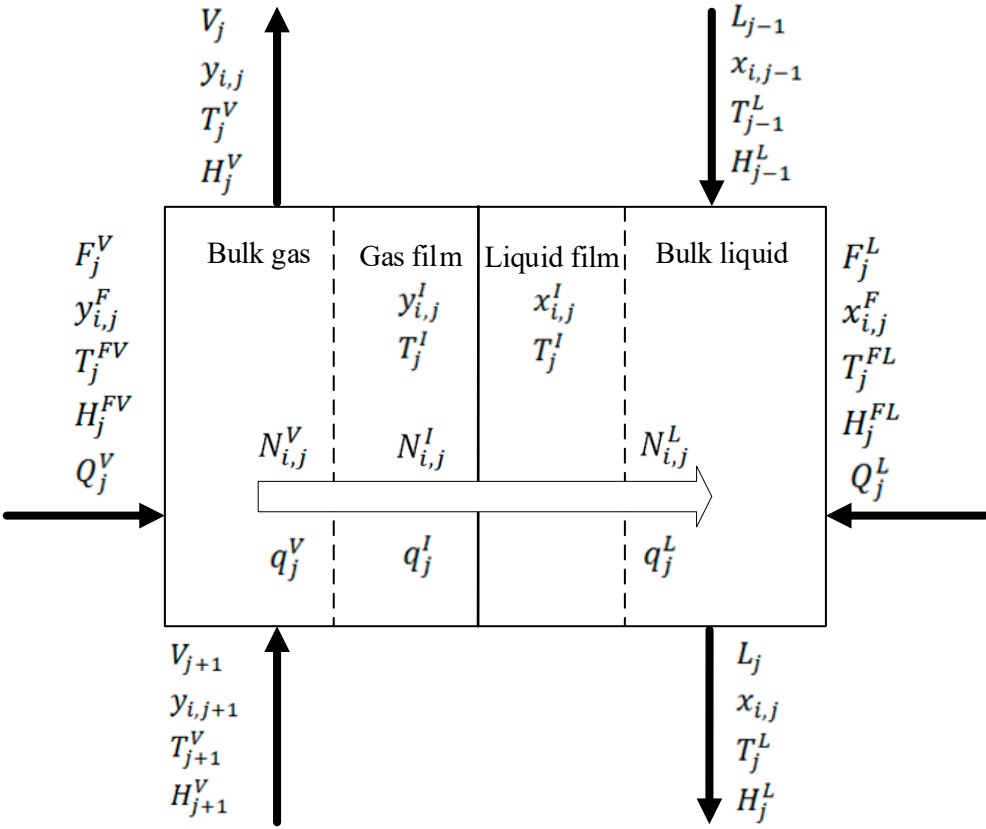

**Figure 4.** Rate-based stage model (reproduced from the ASPEN PLUS software manual).

The material and energy balances around a stage are conducted by applying the MERSHQ equations presented by Taylor and Krishna (1993) [20] as follows:

$$\text{Material balance for bulk vapor}: F_j^V y_{i,j}^F + V_{j+1} y_{i,j+1} + N_{i,j}^V + r_{i,j}^V - V_j y_{i,j} = 0 \tag{4}$$

$$\text{Material balance for bulk liquid}: F_j^L x_{i,j}^F + L_{j-1} x_{i,j-1} + N_{i,j}^L + r_{i,j}^L - L_j x_{ij} = 0 \tag{5}$$

$$\text{Material balance for vapor film}: N_{i,j}^V + r_{i,j}^{fV} - N_{i,j}^I = 0 \tag{6}$$

$$\text{Material balance for liquid film}: N_{i,j}^I + r_{i,j}^{fL} - N_{i,j}^L = 0 \tag{7}$$

$$\text{Energy balance for bulk vapor}: F_j^V H_j^{FV} + V_{j+1} H_{j+1}^V + Q_j^V - q_j^V - V_j H_j^V = 0 \tag{8}$$

$$\text{Energy balance for bulk liquid}: F_j^L H_j^{FL} + L_{j-1} H_{j-1}^L + Q_j^L + q_j^L - L_j H_j^L = 0 \tag{9}$$

$$\text{Energy balance for vapor film}: q_j^V - q_j^I = 0 \tag{10}$$

$$\text{Energy balance for liquid film}: q_j^I - q_j^L = 0 \tag{11}$$

$$\text{Phase equilibrium at the interface}: y_{i,j}^I - K_{i,j} x_{i,J}^I = 0 \tag{12}$$

$$\text{Summation for bulk vapor}: \sum_{i=1}^{n} y_{i,j} - 1 = 0 \tag{13}$$

$$\text{Summation for bulk liquid}: \sum_{i=1}^{n} x_{i,j} - 1 = 0 \tag{14}$$

$$\text{Summation for vapor film}: \sum_{i=1}^{n} y_{i,j}^I - 1 = 0 \tag{15}$$

$$\text{Summation for liquid film}: \sum_{i=1}^{n} x_{i,j}^{I} - 1 = 0 \tag{16}$$

where $F$ is the molar flow rate of feed, $L$ and $V$ are the molar flow rate of liquid and vapor, respectively, $N$ molar transfer rate, $K$ is equilibrium ratio, $r$ is reaction rate, $H$ is enthalpy, $Q$ is heat input to a stage, $q$ is heat transfer rate, $x_i$, $y_i$ are mole fraction of component $i$ in liquid and vapor phases, respectively.

The empirical correlation, according to AICHE (1958) [21], is used for the estimation of the binary mass transfer coefficient for the liquid phase, where Chan and Fair's (1984) [22] correlation is applied for the estimation of the binary mass transfer coefficient for the gas phase. For the estimation of the interfacial area, Zuiderweg's (1982) [23] correlation is used.

In the following, empirical correlations are used in the rate-based model for the estimation of binary mass transfer coefficients and interfacial area $a^I$ (AICHE 1958) [21], (Chan and Fair 1984) [22], (Zuiderweg 1982) [23], (Švandová 2011) [24].

Binary mass transfer coefficient for the liquid

$$k_{i,k}^{L} = \frac{\left(4.127 * 10^{8} D_{i,k}^{L}\right)^{0.5} (0.21313 F_s + 0.15) L t_L}{\overline{\rho}^{L} a^{I}} \tag{17}$$

Binary mass transfer coefficient for the gas

$$k_{i,k}^{V} = \left(10300 - 8670 F_f\right) F_f \left(D_{i,k}^{V}\right)^{0.5} \left(\frac{1-a}{a}\right) \frac{h_{cl}^{0.5} A_b}{a^{I}} \tag{18}$$

Interfacial area

$$a^{I} = \frac{40 A_b}{\phi^{0.3}} \left[ \frac{\left(u_s^{V}\right)^2 \rho_t^{V} h_{cl} FP}{\sigma} \right]^{0.37} \tag{19}$$

For Spray regime where $FP \leq 3.0 l_w h_{cl} / A_b$,
Interfacial area

$$a^{I} = \frac{43 A_b}{\phi^{0.3}} \left[ \frac{\left(u_s^{V}\right)^2 \rho_t^{V} h_{cl} FP}{\sigma} \right]^{0.53} \tag{20}$$

For a mixed froth-emulsion regime where $FP > 3.0 l_w h_{cl} / A_b$ Superficial F-factor

$$F_s = u_s^{V} \left(\rho_t^{V}\right)^{0.5} \tag{21}$$

The superficial velocity of vapor

$$u_s^{V} = \frac{Q_V}{A_b} \tag{22}$$

Fractional approach to flooding

$$F_f = \frac{u_s^{V}}{u_{sf}^{V}} \tag{23}$$

Clear liquid height

$$h_{cl} = 0.6 h_w^{0.5} \left(FP \frac{P A_b}{l_w N_p}\right)^{0.25} \tag{24}$$

Flow parameter

$$FP = \frac{Q_L}{Q_V} \left[\frac{\rho_t^{L}}{\rho_t^{V}}\right]^{0.5} \tag{25}$$

where $k_{i,K}^{L}$ is the binary mass transfer coefficient for the liquid which is predicted by using the AICHE 1958 correlation [21]; $k_{i,K}^{V}$ is the binary mass transfer coefficient for the vapor that is predicted by using

Chan and Fair's (1984) correlation [22]; $D_{i,k}^L$, $D_{i,k}^V$ are diffusivity of the liquid and vapor, respectively; Fs is the superficial F-factor; $L$ is the total molar flow rate of the liquid; $t_L$ is the average residence time for the liquid (per pass); $\overline{\rho}^L$ is the molar density of the liquid; $a^I$ is the total interfacial area for mass transfer that is calculated according to the Zuiderweg's (1982) correlation [23] ; $F_f$ is a fractional approach to flooding; $a$ is the relative froth density ; $h_{cl}$ is the clear liquid height; $A_b$ is the total active bubbling area on the tray; $u_s^V$ is the superficial velocity of vapor; $\rho_t^L$, $\rho_t^V$ are the density of the liquid and vapor, respectively; $FP$ is the flow parameter; $\varnothing$ is the fractional hole area per unit bubbling area; $\sigma$ liquid surface tension; $l_w$ is the average weir length (per liquid pass); $Q_L$, $Q_V$ are volumetric flow rate for the liquid and vapor, respectively; $u_{sf}^V$ is the superficial velocity of vapor at flooding; $l_w$ is average weir height; P is sieve tray hole pitch; $N_p$ is the number of liquid flows.

The heat transfer through the interfacial area is estimated using the Chilton–Colburn analogy. The heat transfer coefficient $h$ is calculated as follows (Simon, Elias et al. 2011) [25]:

$$h = k_G \left( \frac{\rho_G \left( c_p / M_{w,L} \right) \lambda^2}{D^2} \right)^{1/3} \tag{26}$$

where $k_G$ is the mass transfer coefficient for the gas phase, $\rho_G$ is the density of the gas, $c_p$ is specific molar heat capacity, $M_{w,L}$ is the molecular weight of the liquid phase, $\lambda$ is thermal conductivity, and $D$ is the diffusion coefficient.

### 3.2. Tray Column Simulation

For simulating the tray absorber column, ASPEN PLUS software was used due to its extensive property databanks and rigorous equation solvers. The rate-based model was applied due to its empirical correlations for the estimation of mass transfers. For building the rate-based model by using Aspen PLUS, the column was divided into five stages or trays. The inlet gas consists of air and $CO_2$; the mole fraction of $CO_2$ was 0.3. The water was used as an absorbent; the inlet volumetric flow of water was 0.17 m³/h. Two series of the simulation were run: one at 15 Nm³/h of inlet gas flow rate and the other at 20 Nm³/h. The pressure was increased in the range of 0.2–0.3 MPa. The AICHE (1958) correlation [21] was applied for the prediction of the binary mass transfer coefficient for the liquid. Chan and Fair's (1984) correlation [22] was used to predict the binary mass transfer coefficient for the gas. The correlation of Chilton–Colburn was applied to estimate the heat transfer coefficient. The nonrandom two-liquid model (NRTL) was applied for the prediction of phase behavior properties.

Assumptions for the rate-based model approach are summarised below:

- The absorption column is assumed to be adiabatic.
- There is diffusion resistance in film.
- There are no reactions in the film.
- The liquid is well mixed, and the vapor is a plug flow; changes in concentration and temperature in the radial direction are negligible.
- The interfacial surface area is identical for both heat and mass transfer.
- The interface temperature is identical to the bulk liquid temperature since the liquid-side heat transfer resistance is small and balanced to the gas phase (Afkhamipour and Mofarahi 2013) [19].
- Both liquid and gas phases are formally handled as non-ideal mixtures.

## 4. Results and Discussion

### 4.1. Model Validation

The absorber test rig is operated at specified conditions for 15 min, which gives results in time-dependent values for each measured parameter (i.e., temperature, pressure, and gas concentrations). The standard deviation that displays the amount of variation of each measured

parameter is then calculated to estimate the random error. The systematic error of the measurement devices is constant throughout the experiments and is, therefore, not additionally shown in this chapter. Generally, the measurement uncertainty of directly measured values (e.g., pressure, temperature, and flue gas concentrations) depends only on the relative uncertainty of the measuring devices and is given by the relative error. For indirectly measured parameters or calculated values (e.g., volumetric flow rate, where the pressure difference and temperature are applied for its calculation), the Gaussian error propagation method is applied, with the assumption of normally distributed uncertainties. In this work, the volumetric concentrations are detected using the gas analysis unit, and the maximum relative error for $CO_2$ in the different process streams is approximately 3%.

Figure 5 shows a comparison between the outlet volume fraction of $CO_2$ obtained by the experimental model (including the random error bars) and by the Aspen PLUS simulation model as a function of pressure at the inlet gas flow rates of 15 and 20 Nm$^3$/h. It can be noted from Figure 5 that there is an agreement between the simulation results and experimental data at 15 Nm$^3$/h. There is good agreement between the results up to 0.24 MPa at an inlet flow rate 20 Nm$^3$/h; the deviation between the results increase when the pressure is increased between 0.24–0.3 MPa at the inlet gas flow of 20 Nm$^3$/h.

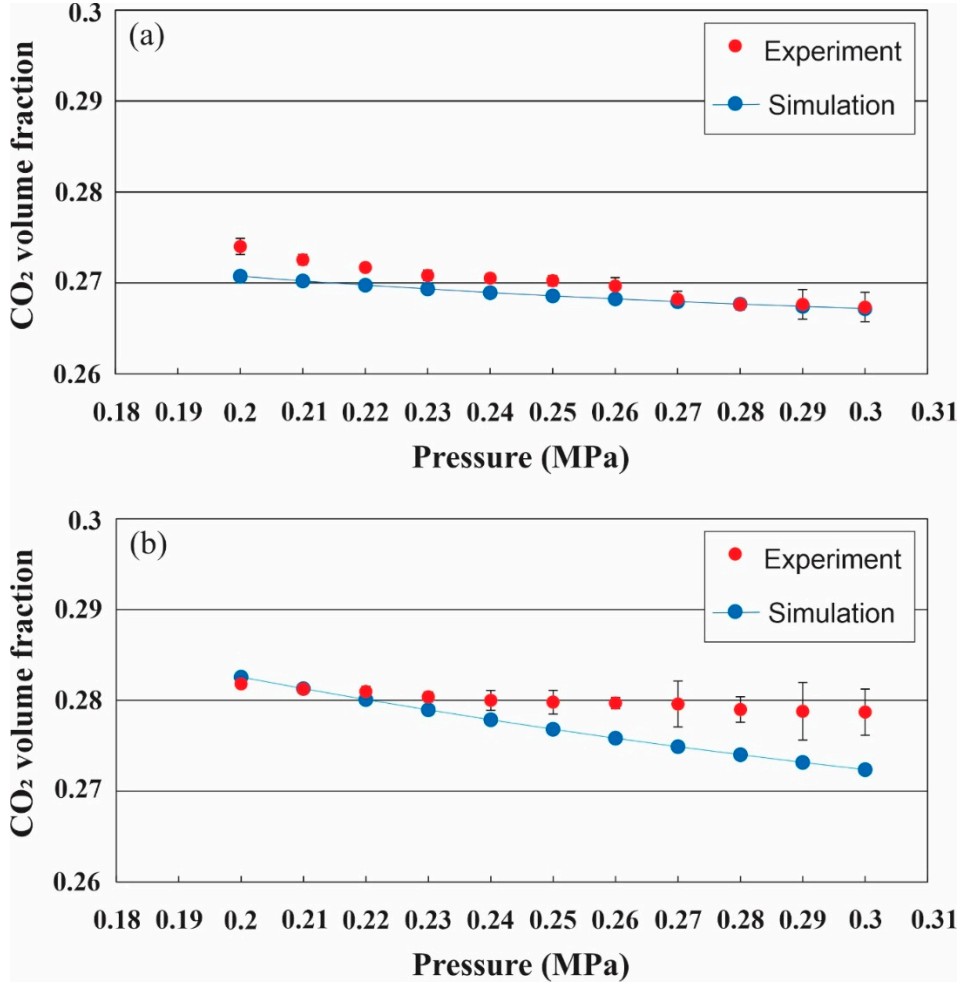

**Figure 5.** Effect of pressure on the outlet volume fraction of $CO_2$ at the inlet gas flow rate of 15 (**a**) and 20 Nm$^3$/h (**b**). The error bars are obtained by the standard errors of measurement.

For assessing the accuracy of the rate-based simulation model, the relative error is calculated between the concentrations estimated using the rate-based simulation model and those obtained from the experimental model. The relative error values are shown in Table 2. As can be seen in Table 2,

there is good agreement between the experimental model and the Aspen PLUS simulation model obtained, with a maximum relative error of 2.27%.

**Table 2.** The relative error between the values calculated using the rate-based model and those obtained from empirical measurement at inlet gas flow rates of 15 and 20 Nm³/h.

| Pressure (MPa) | Inlet Gas Flow Rate (15 Nm³/h) | Inlet Gas Flow Rate (20 Nm³/h) |
|:---:|:---:|:---:|
| 0.2 | 1.195% | 0.262% |
| 0.21 | 0.856% | 0.018% |
| 0.22 | 0.711% | 0.308% |
| 0.23 | 0.551% | 0.511% |
| 0.24 | 0.591% | 0.764% |
| 0.25 | 0.6182% | 1.063% |
| 0.26 | 0.528% | 1.379% |
| 0.27 | 0.104% | 1.678% |
| 0.28 | 0.008% | 1.785% |
| 0.29 | 0.085% | 2.017% |
| 0.3 | 0.062% | 2.271% |

### 4.2. Estimation of the Gas/Liquid Interfacial Area

It was observed during the performance of the experiments that the foam created above trays decreased when the pressure was increased. The effect of pressure increase on the reduction of the foam increased when the inlet gas flow rate was increased. For the estimation of the gas/liquid interfacial area, the validated rate-based model was used. Figure 6 shows the estimated gas/liquid interfacial area calculated by Zuiderweg's (1982) correlation [23] when the pressure is increased. As can be seen in Figure 6, at 15 Nm³/h of the inlet gas flow rate, the gas/liquid interfacial area distinctly decreases when pressure is increased up 0.23 MPa, whereas the gas/liquid interfacial area slightly decreases when the pressure is increased between 0.23–0.3 MPa. In contrast, the gas/liquid interfacial area significantly decreases when pressure is increased between 0.2–0.3 MPa at 20 Nm³/h of the inlet gas flow rate. The trend may be explained when consideration is taken of the fact that the gas/liquid interfacial area depends on superficial velocity, according to Zuiderweg's (1982) correlation [23]. The gas/liquid interfacial area increases when the superficial velocity of the gas is increased, and vice versa. With a constant inlet mass flow rate of the gas, the superficial velocity decreases when pressure is increased (Benadda 1996) [12], which leads to a decrease in the interfacial area according to Zuiderweg (1982) [23].

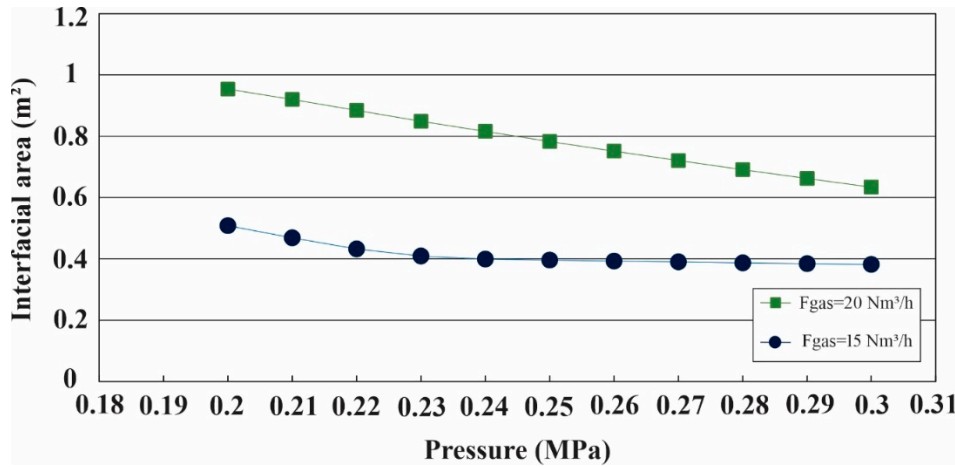

**Figure 6.** Effect of the pressure on the gas/liquid interfacial area at inlet gas flow rates of 15 and 20 Nm³/h.

### 4.3. Studying the Effect of Pressure on the Performance of the Absorber

The performance of the absorber for $CO_2$ capture was measured by estimation of the absorbed rate of $CO_2$ per unit $Nm^3/h$ of the inlet flow rate of $CO_2$. The absorbed rate $N_{CO2}$ of $CO_2$ per unit $Nm^3/h$ of inlet $CO_2$ was calculated using the following equation:

$$N_{CO_2}\left[Nm^3/Nm^3\right] = \left[\left(y_{CO_2,in} - y_{CO_2,out}\right)F_{gas,in}\left[Nm^3/h\right]\right]/F_{CO_2,\ in}\left[Nm^3/h\right] \tag{27}$$

where $F_{gas,in}$ is the inlet gas flow rate, $F_{CO2,in}$ is the inlet $CO_2$ flow rate, $y_{CO2,out}$ is the outlet volumetric fraction of $CO_2$ that was measured by the gas analysis unit, and $y_{Co2,in}$ is the inlet volumetric fraction of $CO_2$, which was calculated as follows:

$$y_{CO_2,in} = \frac{F_{CO_2,in}\left[Nm^3/h\right]}{F_{gas,in}\left[Nm^3/h\right]} = \frac{F_{CO_2,in}\left[Nm^3/h\right]}{F_{CO_2,in}\left[Nm^3/h\right] + F_{air,in}\left[Nm^3/h\right]} \tag{28}$$

$F_{air,in}$ is the inlet air flow rate.

Figure 7 illustrates the effect of the pressure on the absorbed rate of $CO_2$ at 15 and 20 $Nm^3/h$ of inlet gas flow rates. It is clear that at the inlet gas flow rate of 15 $Nm^3/h$, increasing the pressure has a significant effect on the absorbed rate of $CO_2$. The absorbed rate of $CO_2$ increased significantly by increasing the pressure at 15 $Nm^3/h$ of the inlet gas flow rate, whereas at the inlet gas flow rate of 20 $Nm^3/h$, the increase of pressure has a slight effect on the absorbed rate of $CO_2$ compared with the inlet gas flow rate of 15 $Nm^3/h$. Such trends of the curves can be explained when consideration is taken of the influence of pressure on the interfacial area (see Figure 6). It can be seen from Figure 6 that the effect of pressure on decreasing the interfacial area at 20 $Nm^3/h$ is more significant than at 15 $Nm^3/h$ of the inlet gas flow rate. Decreasing the interfacial area at 20 $Nm^3/h$ leads to the reduction of the mass transfer of $CO_2$ from the gas phase to a liquid phase, according to Equations (1)–(3). This explains that there is no distinct increase of the absorbed rate of $CO_2$ per unit $Nm^3/h$ of inlet $CO_2$ although the pressure is increased (as seen in Figure 7). In contrast, at 15 $Nm^3/h$, the absorbed amount of $CO_2$ increased when the pressure is increased since the interfacial area slightly decreases when the pressure is increased. As can be seen in Figure 7, at conditions of 0.2 MPa of pressure and 15 $Nm^3/h$ of inlet gas flow rate, the absorbed rate of $CO_2$ per unit $Nm^3/h$ is 0.087 $\left[Nm^3/Nm^3\right]$. At conditions of 0.3 MPa and 15 $Nm^3/h$, the absorbed rate increases to 0.109 $\left[Nm^3/Nm^3\right]$, with an increased rate of 0.022 $\left[Nm^3/Nm^3MPa\right]$, whereas at conditions 0.2 MPa of pressure and 20 $Nm^3/h$ of inlet gas flow rate, the absorbed rate of $CO_2$ per unit $Nm^3/h$ of inlet $CO_2$ is 0.061 $\left[Nm^3/Nm^3\right]$. At conditions 0.3 MPa and 20 $Nm^3/h$, this absorbed rate increases to 0.071 $\left[Nm^3/Nm^3\right]$, with an increased rate of 0.01 $\left[Nm^3/Nm^3MPa\right]$.

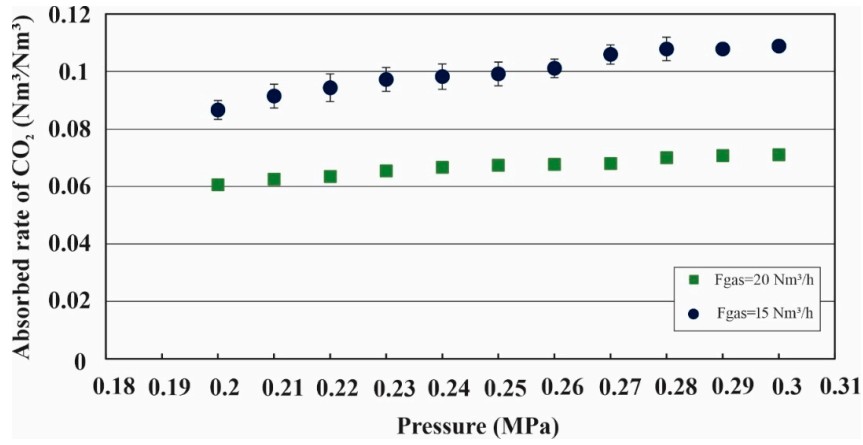

**Figure 7.** Effect of pressure on the absorbed rate of $CO_2$ per unit Nm3/h of inlet $CO_2$ at inlet gas flow rates of 15 and 20 $Nm^3/h$. The error bars are obtained by the standard errors of measurement.

As explained above, one can conclude that the pressure may have an obvious effect on the absorbed rate of $CO_2$ at a high inlet gas flow rate since it decreases the gas/liquid interfacial area, which reflects at the end of the absorbed rate of $CO_2$.

## 5. Conclusions

Our study contributes to the overall knowledge of $CO_2$ absorption under pressure. An absorber test rig was constructed and operated. Furthermore, a rate-based model was built by applying Aspen Plus software to simulate $CO_2$ absorption using water as an absorbent. The model was validated against experimental data at different operating points. The comparison between the results predicted by the rate-based model with the experimental data shows high agreement. The relative error was calculated between the gas concentration calculated using the model and those obtained from the results of the experiment to show the deviation of the model. An analytical study of the $CO_2$ absorption process has also been presented, which highlights the following points:

(1) It shows that the pressure influences the gas/liquid interfacial area. By increasing the pressure, the gas/liquid interfacial area slightly decreases at a low inlet gas flow rate. At a higher inlet gas flow rate, the gas/liquid interfacial area was significantly decreased.
(2) Decreased gas/liquid interfacial area when the pressure is increased has a significant influence on decreasing the absorption rate of $CO_2$.
(3) This study highlights the point of the effect of increasing pressure at a high inlet gas flow rate since it decreases the gas/liquid interfacial area.

This work is a contribution to the knowledge available for modeling studies for $CO_2$ absorption using water as an absorbent in the sieve tray column. Our results confirm other quotes in the literature which are still limited to this issue.

**Author Contributions:** A.A. is responsible for preparing the original draft, developing the applied methodology, and supporting the writing process with his reviews and edits. F.A. conducted the conceptualization and supported the writing process with his reviews and edits. C.H. conducted the conceptualization and supported the writing process with his reviews and edits. B.E. supervised the research progress and the presented work. All authors have read and agreed to the published version of the manuscript.

**Funding:** The authors received no specific funding for this work. The corresponding author would like to thank the Technical University of Darmstadt, enabling the open-access publication of this paper.

**Conflicts of Interest:** The authors declare no conflict of interest.

## Nomenclatures

| | |
|---|---|
| L | molar flow rate of liquid, [kmol/s] |
| V | molar flow rate of vapor [kmol/s] |
| F | molar flow rate of feed, [kmol/s] |
| N | molar transfer rate, [kmol/s] |
| K | equilibrium ratio, [−] |
| r | reaction rate, [kmol/s] |
| H | Enthalpy, [kmol/s] |
| Q | heat input to a stage, [J/s] |
| q | heat transfer rate, [J/s] |
| T | temperature, [K] |
| x | liquid mole fraction [−] |
| y | vapor mole fraction [−] |
| $c_p$ | specific molar heat capacity, [J/kmol K] |
| $M_{w,L}$ | molecular weight of the liquid phase |
| λ | Thermal conductivity, [w/m K] |
| h | Heat transfer coefficient $\left[w/m^2K\right]$ |
| D | diffusion coefficient, $m^2/s$ |

| | |
|---|---|
| $N_{i,G}$ | rate of mass transfer of component i through the gas boundary, [kmol/s] |
| $N_{i,L}$ | rate of mass transfer of component i through the liquid boundary, [kmol/s] |
| $a$ | The interfacial area between gas and liquid phases, m$^2$ |
| $C_{iG}$ | concentration of component i in gas bulk, $\left[\frac{\text{kmol}}{\text{m}^3}\right]$ |
| $C_{iL}$ | concentration of component i in liquid bulk, $\left[\frac{\text{kmol}}{\text{m}^3}\right]$ |
| $C_{iG*}$ | concentration of component i in the interface from the gas side, $\left[\frac{\text{kmol}}{\text{m}^3}\right]$ |
| $C_{iL*}$ | concentration of component i in the interface from the liquid side, $\left[\frac{\text{kmol}}{\text{m}^3}\right]$ |
| $k_{i,k}^V$ | binary mass transfer coefficient for vapor, m/s |
| $k_{i,k}^L$ | binary mass transfer coefficient for liquid, m/s |
| $D_{i,k}^V$ | diffusivity of the vapor, m$^2$/s |
| $D_{i,k}^L$ | diffusivity of the liquid, m$^2$/s |
| $F_s$ | superficial F-factor, $kg^{0.5}/m^{0.5}s$ |
| $t_L$ | average residence time for the liquid (per pass), s |
| $V$ | total molar flow rate of, liquid, vapor, kmol/s |
| $L$ | total molar flow rate of, liquid, vapor, kmol/s |
| $a^I$ | total interfacial area for mass transfer, m$^2$ |
| $\bar{\rho}^V$ | molar density of vapor, kmol/m$^3$ |
| $\bar{\rho}^L$ | molar density of liquid kmol/m$^3$ |
| $F_f$ | fractional approach to flooding, - |
| $a$ | relative froth density, -, Equation (18) |
| $A_b$ | total active bubbling area on the tray, m$^2$ |
| $u_s^V$ | superficial velocity for the vapor, m/s |
| $u_s^L$ | superficial velocity for the liquid, m/s |
| $u_{sf}^V$ | superficial velocity of vapor at flooding, m/s |
| $\rho_t^V$ | density of the vapor, kg/m$^3$ |
| $\rho_t^L$ | density of the liquid, kg/m$^3$ |
| $l_w$ | average weir length (per liquid pass), m |
| $Q_V$ | volumetric flow rate for the liquid, vapor, m$^3$/s |
| $Q_L$ | volumetric flow rate for the liquid, vapor, m$^3$/s |
| $\sigma$ | liquid surface tension N/m |
| $FP$ | flow parameter, - |
| $\varnothing$ | fractional hole area per unit bubbling area on the tray, - |
| $h_{cl}$ | clear liquid height, m |
| $h_w$ | average weir height, m |
| $A_b$ | total active bubbling area on the tray, m$^2$ |
| $N_p$ | number of liquid flow passes, - |
| P | sieve tray hole pitch, m, Equation (24) |

**Subscripts**

| | |
|---|---|
| L | liquid |
| V | vapor |
| F | feed |
| G | gas |
| m$^3$/h | cubic meter per hour |
| i | component |
| n | number of components |
| I | interface |
| f | film |
| j | stage number |

**Abbreviations**

| | |
|---|---|
| NRTL | non-random two-liquid model |
| Nm3/h | cubic meter of gas per hour at the normal temperature and pressure |

| Aspen PLUS | simulation software program |
| MERSHQ | equations of material, energy balances, rate of mass and heat transfer, summation of composition, hydraulic, and equilibrium |
| MEA | 2-aminoethano |
| DEA | 2,2′-iminodiethanol |
| MDEA | N-methyl-2,2′-iminodiethanol |
| IPMEA | 2-(isopropylamino) ethanol |
| TBMEA | 2-(tert-butylamino) ethanol |
| PID controller | proportional–integral–derivative controller |
| MFC | mass flow controller |

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
