# Peer review of "Influence of Pressure on Gas/Liquid Interfacial Area in a Tray Column"

_applsci, doi:10.3390/app10134617_

Round 1

Reviewer 1 Report

The authors studied the effect of pressure on the interface of a simulated gas and water as the liquid in their manuscript entitled “Investigation the effect of the pressure on gas/liquid 2 interfacial area in a tray absorber test rig”. Ths manuscript could appeal to researchers working in the field of CO2 absorption. The following points could be considered before publication.

  1. The introduction section could be improved by the inclusion of aspects like pros and cons of materials used for CO2 absorption in industries.
  2. The unit for the pressure of the absorber is indicated as bara. Please check if it needs to be bar.
  3. In Table 1. Some numerals are bold while others are non-bold. This could be made consistent.
  4. It might be worthwhile to include the details of how the gas/liquid interface area was calculated using the rate-based model.
  5. There is more decrease in initial CO2 volume fraction at a flow rate of 10 Nm3/h than 15 Nm3/h. However, the pressure has less influence at high pressures where the CO2 volume fraction almost closes to 0.26 in both cases.
  6. The trend in the gas/liquid interface area vs the pressure at the two flow rates (Figure 4) does not correlate with CO2 volume fraction data in Figure 3.
  7. English and grammar need to be checked thoroughly as there are several flaws which makes understanding the contents harder.
  8. The references list could be updated by the inclusion of more recent references.

Author Response

Dear Sir or Madam,
Thank you very much for your comments and suggestions, they were very helpful to improve our article. Please see attachments related to a major revision of our article.
Here are some quotes:
- All the comments and suggestions have been processed.
- It was changed of inlet flow rates of inlet gas to be 15 Nm3/h and 20 Nm3/h, since we have found that it is better to study the effect of pressure at high inlet  gas flow rates. The model has been updated according to new experiments.- - -  - Comments and suggestions:
1- The introduction section could be improved by the inclusion of aspects like pros and cons of materials used for CO2 absorption in industries.
It has been done, please see the Line 28 in (Page 1).

2- The unit for the pressure of the absorber is indicated as bara. Please check if it needs to be bar.
It has been processed, the unit for the pressure has been changed to SI Unit (MPa).

3- In Table 1. Some numerals are bold while others are non-bold. This could be made consistent.
It has been processed.

4- It might be worthwhile to include the details of how the gas/liquid interface area was calculated using the rate-based model.
It has been done, please see the part; Rate-based modelling for CO2 absorption, (Page 6).

5- There is more decrease in initial CO2 volume fraction at a flow rate of 10 Nm3/h than 15 Nm3/h. However, the pressure has less influence at high pressures where the CO2 volume fraction almost closes to 0.26 in both cases.
Please see new results at 15 and 20 Nm3/h, if there are any comments, please do not hesitate to write to us.

6- The trend in the gas/liquid interface area vs the pressure at the two flow rates (Figure 4) does not correlate with CO2 volume fraction data in Figure 3.
It has been found that the pressure has a passive influence on the co2 absorption at high inlet gas flow rate. We have been observed during performing the experiments that the height of foam on the tray (which consider as part of interfacial area been the gas and liquid) decreases
when pressure is increased, decreasing of foam or the interfacial area leads to decrease the mass transfer been the gas and the liquid, so one can see from Figure 5 that the outlet volume fraction of CO2 at 15 nm3/h decreased significantly when the pressure is increased because the interfacial area decreased slightly. In contrast, at 20 Nm3/h, the volume fraction of the Co2 decreased slightly when pressure is increased because the interfacial area decreased significantly (Please see Figure 6)

7- English and grammar need to be checked thoroughly as there are several flaws which makes understanding the contents harder.
It has been processed.

8- The references list could be updated by the inclusion of more recent references.
It has been done.

If you have any comments and suggestions please do not hesitate to contact us.

Thank you very much and best regards,
Adel Almoslh

Reviewer 2 Report

The authors have investigated the effect of the pressure on gas/liquid interfacial area in a tray absorber test rig. Overall the paper is well written and structured. The manuscript needs to be revised before it could be considered for publication.

  1. English editing is needed since there are numerous grammatical and typo errors, e.g., the title of the manuscript.
  2. The literature review is not sufficient. The authors should include more papers on the topic "pressure effect on the interfacial area". The references are not sufficient to back the research topic. This area must be improved.
  3. The resolution and labeling of Figure 2 are weak and not visible.
  4. The absolute units 'bara' is generally deprecated. It is recommended to use other SI units of pressure for a better understanding of readers.
  5. A detailed description of the CFD (RateFrac model) with equations and boundary conditions must be presented.
  6. What is the rate-based model, as mentioned in Line 181 (Page 7)? The rate-based model's description must be included. How was it built, as mentioned in conclusion?
  7. The authors should provide the reason that the interfacial area for 15 Nm3/h shows a sudden decrease with increasing pressure and then become almost constant above 2.8 bara.
  8. With a 15 Nm3/h gas flow rate, increasing the pressure has minimal effect on absorbed CO2, and it became almost constant above 2.9 bara (Fig. 5). Also, the trend of absorbed CO2 with 10 Nm3/h is steeper than those with 15 Nm3/h.The authors should provide the reason.

Author Response

Dear Sir or Madam,

Thank you very much for your comments and suggestions, they were very helpful to improve our article. Please see attachments related to a major revision of our article.

Here are some quotes:

- All the comments and suggestions have been processed.

- It was changed of inlet flow rates of inlet gas to be 15 Nm3/h and 20 Nm3/h, since we have found that it is better to study the effect of pressure at high inlet gas flow rates. The model has been updated according to new experiments.- - - - Comments and suggestions:

1- English editing is needed since there are numerous grammatical and typo errors, e.g., the title of the manuscript.

It has been done.

2- The literature review is not sufficient. The authors should include more papers on the topic "pressure effect on the interfacial area". The references are not sufficient to back the research topic. This area must be improved.

It has been done.

3- The resolution and labeling of Figure 2 are weak and not visible.

It has been processed.

4- The absolute units 'bara' is generally deprecated. It is recommended to use other SI units of pressure for a better understanding of readers.

It has been processed, the unit for the pressure has been changed to SI Unit (MPa).

5- A detailed description of the CFD (RateFrac model) with equations and boundary conditions must be presented.

It has been done, please see the part; Tray column simulation, (Page 9).

6- What is the rate-based model, as mentioned in Line 181 (Page 7)? The rate-based model's description must be included. How was it built, as mentioned in conclusion?

 It has been done, please see the part; Rate-based modelling for CO2 absorption in (Page 6), and part Tray column simulation in (page 9).

7- The authors should provide the reason that the interfacial area for 15 Nm3/h shows a sudden decrease with increasing pressure and then become almost constant above 2.8 bara.

Please see new results at 15 and 20 nm2, if there are any comments, please do not hesitate to write to us.

8- With a 15 Nm3/h gas flow rate, increasing the pressure has minimal effect on absorbed CO2, and it became almost constant above 2.9 bara (Fig. 5). Also, the trend of absorbed CO2 with 10 Nm3/h is steeper than those with 15 Nm3/h.The authors should provide the reason.

It has been done according the new results, please see line 292 in page(12).

If you have any comments and suggestions, please do not hesitate to contact us.

Thank you very much and best regards,

Adel Almoslh

Round 2

Reviewer 2 Report

The reviewer will have appreciated if the authors have highlighted the modifications in the revised manuscript since it is difficult to find the modified part in the present form.

It seems that the authors have addressed the majority of the comments.

However, the authors have not improved the literature review by adding more articles related to the topic. The reviewer believes that literature review and references are not sufficient to back up the research.

Figs.1 and 2 are not up to the standards. Fig 1 must be labeled for a better understanding of the readers. The labels of Fig. 2 are still not clear and readable. These must be improved.

The y-axis of Fig. 5 must be modified. The y-axis (CO2 volume fraction) must start from 0.26. This will enable to compare experiments and simulation effectively. Also discuss the deviation between the results of experiment and simulation.

The experiment must be conducted at least thrice to minimize the relative errors. The error plots must be included in Fig. 5.

Author Response

Dear Sir or Madam,
Thank you very much for your comments and suggestions; they were helpful to improve our article. Please see attachments related to a major revision of our article.
- All the comments and suggestions have been processed; here are some quotes:
1- The article has been improved by adding more articles and references related to the topic, please see the highlighted part of “Studying the effect of pressure on gas/liquid interfacial area” in line 44.
2- Fig 1 has been labelled, and Fig. 2 has been improved very well.
3- The y-axis (CO2 volume fraction) has been changed to be started from 0.26. Also, the deviation between the results of experiment and simulation has been done.
4- All experiments have been already conducted at least thrice. The error have been calculated and added in Fig. 5.
If you have any comments and suggestions, please do not hesitate to write to us.
Thank you very much and best regards,
Adel Almoslh
